

**Prediction of basin-scale river channel migration based on**
**landscape evolution numerical simulation**
Jitian Wu[1], Xiankui Zeng[1], Qihui Wu[1], Dong Wang[1], and Jichun Wu[1]
[1] Key Laboratory of Surficial Geochemistry, Ministry of Education, School of Earth
Sciences and Engineering, Nanjing University, Nanjing 210023, China
**Correspondence:** Xiankui Zeng (xiankuizeng@nju.edu.cn)

8                    Submitted to *Hydrology and Earth System Sciences*



## Abstract

The basin-scale river channel migration, driven by multiple factors such as hydrometeorological conditions, tectonic movements, and human activities, exerts a profound influence on regional morphological features, water resource, and ecosystem over long-term evolution. Conventional river dynamics approaches struggle to quantitatively characterize basin-scale channel migration due to difficulties in incorporating factors like basin hydrological processes and tectonic activities. This study proposed a novel technique for the numerical simulation of river channel migration, integrating a fully coupled multi-processes landscape evolution model (e.g., hydrological, geomorphic and tectonic processes) with channel extraction. Furthermore, to address model parameter uncertainty, a Markov chain Monte Carlo (MCMC) method with a modified likelihood function is used for parameter uncertainty quantification. Simultaneously, a computationally efficient Long Short-Term Memory (LSTM)-based surrogate model for channel migration is developed to overcome the computational bottleneck in uncertainty analysis. Applied to the Kumalake River Basin within China's Tarim Basin, the study employs the Landscape Evolution-Penn State Integrated Hydrologic Model (LE-PIHM) to construct the landscape evolution model. Combined with channel extraction, it simulates historical (2000-2021) and future (2021-2100) landscape evolution and channel migration processes. Results demonstrated that the developed river channel migration model, aided by parameter uncertainty analysis, reliably captures the dynamics of channel migration in the study area during 2000-2021. Additionally, the LSTM-based surrogate model achieves high accuracy, effectively resolving computational challenges in parameter uncertainty analysis. Predictions under different climate scenarios



reveal significant variations in future channel evolution, indicating that climate change will

profoundly reshape basin geomorphic features and river patterns.



## 1. Introduction

Basin-scale river channel migration is the result of interactions among multiple spheres within the complex earth system, influenced by various factors including meteorological, hydrological, and geological conditions (Li et al., 2023; Desormeaux et al., 2021). Over long temporal and basin scales, river channel migration regulates the spatial configuration of river networks, exerting significant impacts on regional water resources, ecological environments, and the development of civilizations. For instance, the substantial downstream migration of the Euphrates River between approximately 2112–2004 BCE contributed to the collapse of the Sumerian civilization (Hritz et al., 2010). The diversion of the lower Tarim River in 630 CE led to the disappearance of the ancient Loulan Kingdom (Yu et al., 2016; Shao et al., 2022). Quantitative research on river channel migration at the basin scale is crucial; it can not only inform projections of water resource distribution under climate change scenarios but also facilitate the reconstruction of linkages among channel evolution, fluvial ecosystems, and the trajectories of human civilizations (Hickin 1983; Zhou et al., 2022; Zhen et al., 2025).

River channel migration in the basin scale involves multiple coupled processes including surface water and groundwater flow, weathering and erosion, and tectonic uplift. These processes operate across broad spatiotemporal scales, exhibit complex mechanistic interactions, and are highly susceptible to anthropogenic disturbance. Numerical modeling serves as the principal approach for quantitatively characterizing these dynamics. Among various modeling strategies, river channel migration models grounded in fluvial dynamics have been widely used. For example, Ikeda et al. (1981) developed a single meander segment model by coupling flow fields with erosion rates. Morón et al. (2017) employed



Delft3D (Lesser et al., 2004) to simulate the evolution of channel segments in the Nile
River, Columbia River, Congo River and Negro River. Hsu et al. (2022) utilized Nays2DH
(Ali et al., 2017) to simulate braided river morphology in the lower Dajia River, and
identified that channel width as a primary factor governing migration direction. However,
these methods generally focus on partial river channel domains (e.g., meander reaches) and
fail to incorporate hydrologic processes and tectonic activities at the basin scale, thus
limiting their applicability to river channel migration over engineering timescales and
channel segment scales.

As a typical geomorphic unit, river channels are fundamentally governed by landscape

evolution processes (Lisenby et al., 2020). Landscape evolution models (LEMs) are
numerical tools designed to quantify elevation changes across watersheds over geological
timescales, incorporating hydrologic processes and tectonic uplift (Bishop, 2007; Tucker
and Hancock, 2010; Hou et al., 2025). By integrating LEMs with river channel extraction
techniques, it becomes feasible to simulate long-term, basin-scale channel migration.
Commonly used LEMs include CASCADE (Braun et al., 1997), CHILD (Tucker et al.,
2001), CAESAR-Lisflood (Coulthard et al., 2013), DAC (Goren et al., 2014; Yang et al.,
2015), Landlab (Barnhart et al.,2020; Litwin et al.,2024), all of which have been widely
used for simulating landscape evolution. Nevertheless, these models often simplify or
neglect groundwater dynamics and lateral erosion processes (Whipple et al., 2017), making
it difficult to accurately capture these crucial hydrological and geomorphic processes in
large-scale, long-term watershed landscape evolution simulations.

Zhang et al. (2016) developed LE-PIHM by coupling surface-subsurface hydrologic

processes with slope and channel sediment transport, while accounting for bedrock



weathering and tectonic uplift, based on the PIHM framework (Qu et al., 2007). LE-PIHM
is particularly suited for quantifying landscape evolution processes over long durations and
basin extents.

Nevertheless, current studies rarely integrate multiple processes for real basin-scale

simulations of coupled landscape evolution and river channel migration. Meanwhile,
LEMs contain a large number of parameters to be identified, the non-negligible parameters
uncertainty can lead to unreliable simulations of landscape evolution and river channel
migration, which has not been adequately addressed in current researches (Temme et al.,
2009; Neuendorf et al., 2018). To quantify river channel migration at the basin scale under
the coupled effects of multiple processes, the LE-PIHM was used to establish a landscape
evolution model in this study, and the distribution of river channels was identified using a
river channel extraction technique. In addition, Long Short-Term Memory (LSTM)
surrogate modeling and Bayesian uncertainty analysis were employed to quantify
parameter uncertainty in the LEM.

This study selects the Kumalake River Basin within China's Xinjiang Tarim Basin as

the research area. This basin features diverse geomorphic types and complex climatic
conditions, and has experienced significant river channel migration in recent decades
(Wang et al., 2024), making it an ideal site for conducting basin-scale simulations of terrain
evolution and river channel dynamics. Based on the identified model parameters
distribution of LE-PIHM, the river channel migration process in the study area over the
past two decades was quantitatively reconstructed. Finally, this study conducted future
scenario simulations of landscape evolution and river channel migration under projected
climate change through the end of the 21st century.




The structure of this paper is as follows: Sect. 2 outlines the methodology and overall
workflow; Sect. 3 introduces the construction of the basin-scale river channel migration
model and the development of an LSTM-based surrogate model for uncertainty analysis;
Sect. 4 presents the results and discussion; and Sect. 5 provides concluding remarks and
summarizes the key findings.

## 2. Methodology

### 2.1 Framework of basin-scale river channel migration prediction

The framework of predicting basin-scale river channel migration in this paper consists
of three parts (Fig.1).
Part 1: Establishment of the basin-scale river channel migration model. The basin-
scale river channel migration model is implemented in two steps. First, landscape evolution
is simulated using LE-PIHM to obtain the elevation distribution of the study area.
Subsequently, the DEM is processed using the D8 algorithm to extract the spatial
distribution of river channel.
Part 2: Development of a LSTM-based surrogate model for efficient parameter
uncertainty analysis. To improve the efficiency of parameter identification, a surrogate
model corresponding to the original river channel migration model (Part 1) is developed
for the reconstruction period (2000–2021). Parameter sensitivity analysis is first conducted
to identify the key landscape evolution parameters of LE-PIHM. Then, 3,000 parameter
sets are sampled and input into the basin-scale river channel migration model (Part 1) to
generate the associated planar channel coordinates, which serve as the training datasets.
The LSTM is trained using these data to construct a surrogate model of basin-scale river
channel migration, substantially reducing the computational burden of parameter





uncertainty analysis.
Part 3: Parameter uncertainty analysis and the prediction of channel migration. Based
on the LSTM-based surrogate model (Part 2), parameter uncertainty analysis is conducted
using a modified-likelihood Markov chain Monte Carlo (MCMC) approach constrained by
observed river channel data from 2000 to 2021. The resulting posterior distribution of river
channel enable the reconstruction of channel migration processes over 2000–2021. The
maximum-likelihood posterior parameter set is then selected, and the original river channel
migration model (Part 1) is executed under future climate scenarios to predict channel
migration from 2021 to 2100.



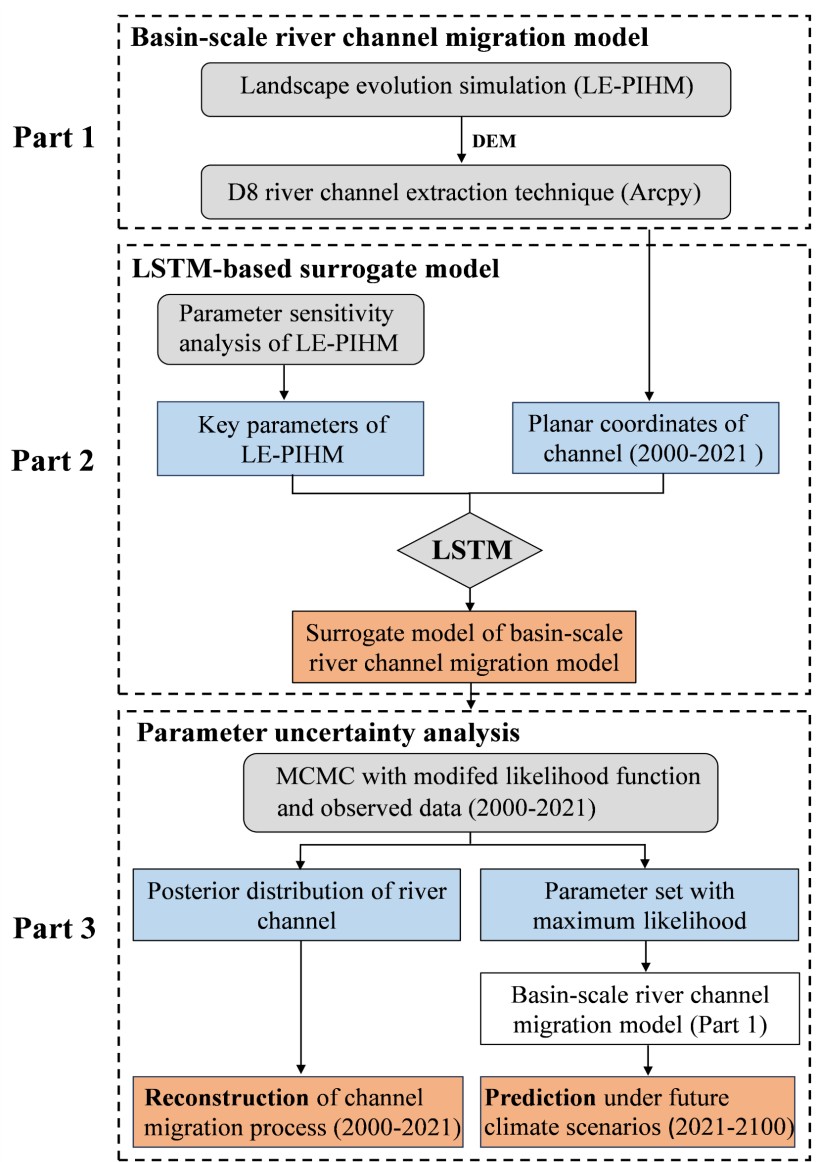

**Figure 1.** Technical roadmap of basin-scale river channel migration prediction
**2.2 Basin-scale river channel migration model**
**2.2.1 Landscape evolution simulation**
The Landscape Evolution–Penn State Integrated Hydrologic Model (LE-PIHM) was
used to simulate the landscape evolution processes. LE-PIHM couples the processes of

 

surface water and groundwater, snow accumulation and melt, hillslope and river channel
sediment transport, weathering and erosion, as well as tectonic uplift (Fig. 2). It is a basin-
scale fully coupled hydrologic-process-based landscape evolution model (Zhang et al.,

2016).

The model simulates surface elevation change based on the principle of mass
continuity. According to the law of mass conservation, the geomorphic process equation
can be expressed as the temporal variation of the mass of the regolith and bedrock.

$$
\begin{aligned}
\frac{\partial(\sigma_{re}h dx dy)}{\partial t} + \frac{\partial(\sigma_{ro}e dx dy)}{\partial t} = {}& \sigma_{re}q_c dy - [\sigma_{re}q_c dy + \frac{\partial(\sigma_{re}q_c dy)}{\partial x}dx] \\
& + \sigma_{re}q_c dx - [\sigma_{re}q_c dx + \frac{\partial(\sigma_{re}q_c dy)}{\partial y}dy] \\
& + \sigma_{re}q_s dy - [\sigma_{re}q_s dx + \frac{\partial(\sigma_{re}q_s dy)}{\partial y}dx] \\
& + \sigma_{re}q_s dx - [\sigma_{re}q_s dx + \frac{\partial(\sigma_{re}q_s dy)}{\partial y}dy] \\
& \sigma_{ro}U dx dy
\end{aligned}
\tag{1}
$$

where $\sigma_{re}$ is the bulk density of regolith (kg/m³); $\sigma_{ro}$ is the bulk density of bedrock (kg/m³);
$h$ is the regolith thickness (m); $e$ is the elevation of the bedrock surface (m); The regolith
thickness $h$ is defined as the difference between the ground surface elevation $z$ and the
bedrock elevation $e$; $q_c$ is the lateral volumetric flux of regolith (m²/yr), driven by processes
such as soil creep; $q_s$ is the surface sediment flux by overland flow (m²/yr); $U$ is the tectonic
uplift rate (m/yr).
The governing equations for hydrologic processes describe the water flux dynamics
from the vegetation canopy to the regolith layer. These processes can be represented as
follows:



$$
\begin{cases}
\dfrac{\partial \psi_{\text{canopy}}}{\partial t} = \upsilon Frac(1-f_s)p - E_c - TF \\[2mm]
\dfrac{\partial \psi_{\text{snow}}}{\partial t} = f_s p - SM \\[2mm]
\dfrac{\partial \psi_{\text{surf}}}{\partial t} = \nabla q_{sw} + p_{\text{net}} - I - E_s \\[2mm]
\dfrac{\partial \psi_{\text{unsat}}}{\partial t} = I - R - E_g - E_{gt} \\[2mm]
\dfrac{\partial \psi_{\text{sat}}}{\partial t} = \nabla q_{gw} + R - E_{sat} - E_{tsat}
\end{cases}
\tag{2}
$$

where: $\Psi_{\text{canopy}}$ is the canopy water storage (m); $\Psi_{\text{snow}}$ is the snow depth (m); $\Psi_{\text{surf}}$ is the
surface water depth (m); $\Psi_{\text{unsat}}$ is the water storage in the unsaturated zone (m); $\Psi_{\text{sat}}$ is
the groundwater (saturated zone) storage (m); $vFrac$ is the fraction of vegetation
coverage; $f_s$ is the fraction of precipitation falling as snow; $P$ is the precipitation rate
(m/day); $E_c$ and $E_s$ are the evaporation rates from the canopy and surface water, respectively
(m/day); $TF$ is the throughfall rate from canopy to ground (m/day); $SM$ is the snowmelt rate
(m/day); $p_{\text{net}}$ is the net precipitation reaching the ground surface (m/day); $I$ is the
infiltration rate (m/day); $E_g$ and $E_{\text{sat}}$ are the evaporation rates from the unsaturated and
saturated zones, respectively (m/day); $E_{gt}$ and $E_{tsat}$ are the transpiration rates from the
unsaturated and saturated zones, respectively (m/day); $q_{sw}$ is the unit-width overland flow
rate (m²/day); $q_{gw}$ is the unit-width groundwater (lateral) flow rate (m²/day).

The values of $q_{sw}$ and $q_{gw}$ are determined by the Manning equation and Darcy's law,

respectively, as follows:

$$
q_{sw} = \frac{\psi_{surf}^{5/3}}{n_s}(\nabla(\psi_{surf}+z)^{1/2})
\tag{3}
$$

$$
q_{gw} = \psi_{sat} k_{sat}\nabla(\psi_{sat}+e)
\tag{4}
$$



where $n_s$ is the Manning's roughness coefficient; $k_{\text{sat}}$ is the horizontal hydraulic
conductivity of the aquifer (m/day). For detailed formulas and variable descriptions, please
refer to Qu and Duffy (2007).

The above governing equations (1-4) couple hydrological processes, hillslope and

channel sediment transport processes, and tectonic movements to form a state-of-the-art
hydro-geomorphic model for simulating landscape evolution.

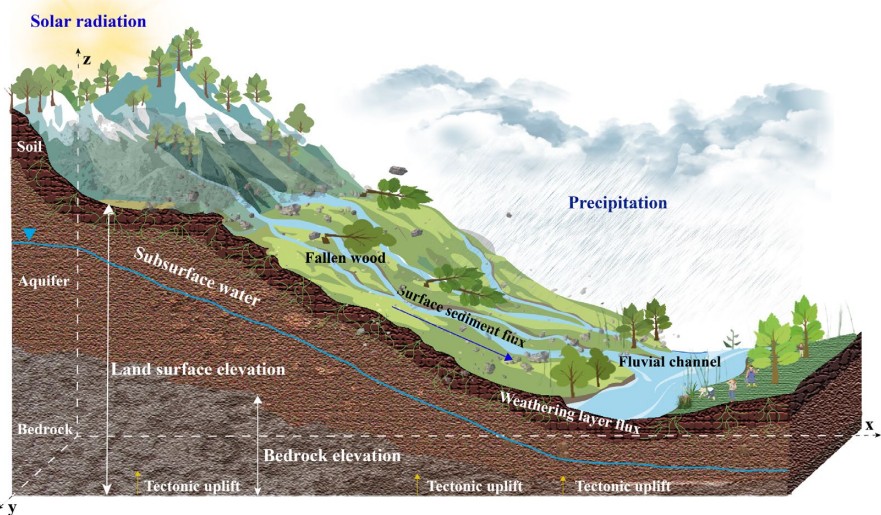


**Figure 2.** Schematic diagram of the Landscape evolution process.
**2.2.2 D8 algorithm river channel extraction technique**

The river channel extraction in basin-scale is implemented through a topography-

based extraction technique. This technique is based on the principle of maximum gradient
and identifies river channels within the watershed using surface elevation data. Specifically,
the landscape evolution simulation provides the surface elevation distribution of the study
area, from which a digital elevation model (DEM) is constructed. The DEM is first filled
to remove depressions, followed by spatial analysis to determine surface flow direction and



compute flow accumulation. River channel distribution is then extracted based on the
accumulated flow.

The standard D8 algorithm, known for its simplicity and practicality, is currently the

most widely used and reliable method for determining water flowpaths from DEM
(O'Callaghan & Mark, 1984; Tarboton, 1997). The ArcPy scripting in ArcGIS, which
implements the standard D8 algorithm (Esri, 2022), will be used in this study to extract
watershed river channels through the ArcGIS hydrological analysis platform.

### 2.2.3 Long Short-Term Memory algorithm

Long Short-Term Memory (LSTM) is a specialized type of Recurrent Neural Network

(RNN), originally proposed by Hochreiter et al. (1997) and later extended and popularized
by Graves et al. (2005). Traditional RNNs often suffer from the vanishing or exploding
gradient problem when processing long sequential data, making it difficult to capture long-
term dependencies effectively. LSTM addresses this issue by introducing gated
mechanisms namely the forget gate, input gate, and output gate, which dynamically
regulate the flow of  information. These gates allow the network to selectively store, update,
and output information within memory cells, thereby effectively addressing the gradient
instability problem in long-sequence modeling.

To address the computational burden caused by repeated model evaluations during

the parameter uncertainty analysis of the basin-scale river channel migration model, this
study constructs a surrogate model using an LSTM network. Specifically, the LSTM
algorithm is employed to build the nonlinear response relationship between the key
parameters of LE-PIHM and the spatial distribution of river channels (i.e., planar
coordinates of reaches) within the study area.





### 214    2.3 Bayesian uncertainty analysis

### 215    2.3.1 Markov chain Monte Carlo simulation

Markov chain Monte Carlo (MCMC) is a statistical simulation technique based on
Bayesian theory. Its core idea is to construct a Markov chain that iteratively explores the
parameter probability space to generate samples from the target posterior distribution. As
the chain evolves, its stationary distribution converges to the posterior distribution of the
parameters of interest (Vrugt et al., 2009).
MCMC integrates observational data through Bayes' theorem, enabling parameter
samples to progressively converge from the prior distribution $p(\theta)$ to the posterior
distribution $p(\theta \,|\, D)$.
$$p(\theta \,|\, D) = \frac{L(\theta \,|\, \mathrm{D})\,p(\theta)}{\int L(\theta \,|\, \mathrm{D})P(\theta)d(\theta)} \tag{5}$$

where $L(\theta \,|\, \mathrm{D})$ represents the likelihood function of a parameter sample $\theta$, D represents
the observed data. The likelihood function $L$ is typically defined as a Gaussian likelihood
function:
$$L(\theta^i \,|\, \mathrm{D}) = \frac{1}{2\pi^{n/2}\,|\Sigma|^{1/2}}\exp\left[-\frac{[\mathrm{D}-f(\theta^i)]^{\mathrm{T}}\,\Sigma^{-1}[\mathrm{D}-f(\theta^i)]}{2}\right] \tag{6}$$

where n is the number of observed data points, $f(\theta^i)$ denotes the hydrologic model
simulation result given the parameter $\theta^i$, and $\Sigma$ is the covariance matrix of the simulation
residuals.

### 232    2.3.2 Hausdorff distance and the modified likelihood function

This study uses the average Hausdorff distance as a metric to quantify the discrepancy
between the simulated river channel and the real river channel. The Hausdorff distance is
an effective tool for assessing positional differences between two curves (Schütze et al.,



2012), and is suitable for quantitatively evaluating the spatial deviation associated with
river channel migration.

The core concept of the Hausdorff distance is to treat two curves as two sets of discrete

points. Suppose we have two point-sets, $A = \{a_1,\ldots, a_p\}$, $B = \{b_1,\ldots, b_q\}$, p and q represent
the number of points in sets A and B, respectively, the bidirectional Hausdorff distance
$HD$(A, B) between sets A and B is defined as:

$HD$(A, B) = Max $[h$(A, B)$, h$(B, A)$]$                     (7)

where $h$(A, B) and $h$(B, A) represent the one-sided Hausdorff distances. Specifically, $h$(A,
B) denotes the set of minimum distances from each point in set A to the nearest point in
set B, $h$(A, B) = $h(a_{1min}, a_{2min}, \ldots, a_{pmin})$, where $a_{1min}$ is the minimum distance from point
$a_1$ to the points in set B. Similarly, $h$(B, A) denotes the set of minimum distances from each
point in B to the nearest point in A.

In this study, we modified Eq. (7) by replacing the maximum operation in the one-

sided distance with the mean of the minimum distances, and set p equal to q. This
modification better captures the overall spatial discrepancy between two river channel
curves and is referred to as the average Hausdorff distance ($H$). A smaller value of $H$
indicates that the simulated river channel more closely matches the observed channel.
$$H = \frac{1}{2p}\sum_{i=1}^{p}[h(A,B)+h(B,A)] \tag{8}$$

In the uncertainty analysis of river channel migration simulation, the likelihood

function quantifies the degree of fit between the simulated and observed river channels. To
enable the parameter uncertainty quantification through MCMC, the original likelihood
function (i.e., Eq. (6)) is revised by treating the average Hausdorff distance ($H$) as the



simulation target. The observed value of $H$ (denoted as $H_{obs}$) is set to 0, this indicates the
$H$ between the real river channel and itself. Thus, the residuals in the likelihood function
can be expressed as:
$$H_{obs} - H = 0 - H = -H \qquad (9)$$

combining Equation (6) with Equation (9) and taking the natural logarithm, a modified
likelihood function designed for quantifying river channel simulation is obtained:
$$\ln L = -\frac{1}{2}[\frac{H^2}{\Sigma^2} + \ln(2\pi\Sigma^2)] \qquad (10)$$

In other words, this modified form of the likelihood function (Equation 10) is used in the
MCMC-based uncertainty analysis of river channel model parameters.
**3. Construction of the river channel migration model**
**3.1 Study area**

The Kumalake River Basin is located in the northwestern corner of the Tarim Basin

in Xinjiang, China, covering an area of about 3,500 square kilometers. The basin is
bordered by the towering Tianshan mountains to the north, the flat Aksu Plain to the south,
and the Toxkan River Basin to the west. The landscape exhibits a distinct elevation gradient
sloping from north to south, with a complex geomorphic setting comprising mountainous
hillslopes, valley plains, and fluvial terraces (Fig. 3). The Kumalake River, approximately
89.34 km in length, is the largest tributary of the Aksu River. It flows from the northwest
to the southeast across the study area and exits at the southeastern edge of the basin, where
it joins the Tuoshigan River. The combined flow continues into the Aksu River and
eventually discharges into the main stem of the Tarim River (Tang et al., 2007).



279   The Kumalake River has experienced pronounced channel migration during the last

280 two decades. This study selects the Kumalake River Basin as the case study area, with a

281 focus on simulating river channel migration over the period from 2000 to 2021. The

282 simulation is conducted within the landscape evolution modeling framework, and a

283 parameter uncertainty analysis is performed to improve the reliability of the model outputs.

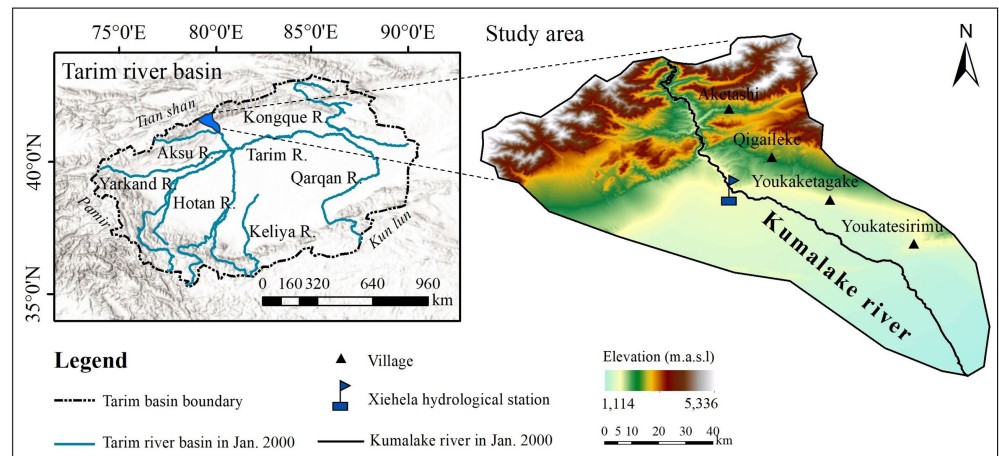

285 **Figure 3.** Schematic diagram of the watershed in the study area. Basemap: Esri World

286 Hillshade (Esri).

## 3.2 Model input data

288   In this study, LE-PIHM combined with a river channel extraction technique is

289 employed to simulate the spatial distribution of river channels. Notably, four categories of

290 physical properties within the study area, i.e., soil, aquifer, bedrock, and land cover, exhibit

291 significant spatial heterogeneity (see Table 1 and Fig. 4). The driving force data in LE-

292 PIHM primarily include leaf area index, precipitation rate, air temperature, downward

293 shortwave radiation, snowmelt rate, wind speed, and relative humidity (Table 1)

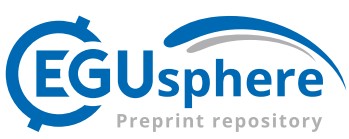

**Table 1.** Types and sources of input data for the landscape evolution model.

| Data Type | Spatial Resolution | Temporal Resolution | Data Source | URL |
|---|---|---|---|---|
| Soil type | 1:5000000 | / | Food and Agriculture Organization of the UN | https://data.apps.fao.org/?lang=en |
| Bedrock type | 1:10000000 | / | International Soil Reference and Information Center | https://soilgrids.org/ |
| Land cover | 10 m | / | Environmental Systems Research Institute, Inc | https://livingatlas.arcgis.com/landcover/ |
| Aquifer type | / | / | China Cartographic Publishing House | https://www.sinomaps.com/pub/ |
| Leaf area index | 0.5°×0.625° | 1h | | |
| Surface roughness | 0.5°×0.625° | 1h | | |
| Precipitation rate | 0.25° | 3h | | |
| Air temperature | 0.25° | 3h | | |
| Shortwave radiation | 0.25° | 3h | | |
| Canopy interception storage | 0.25° | 3h | National Aeronautics and Space Administration | https://www.nasa.gov/ |
| Soil water storage | 0.25° | 3h | | |
| Snow depth | 0.25° | 3h | | |
| Snow melt rate | 0.25° | 3h | | |
| Wind speed | 0.25° | 3h | | |
| Relative humidity | 1.25° | 3h | | |
| Initial surface elevation | 30 m | / | | |
| Initial bedrock depth | 250 m | / | International Soil Reference and Information Center | https://isric.org/ |

All links in the table were last accessed on 2 December 2025



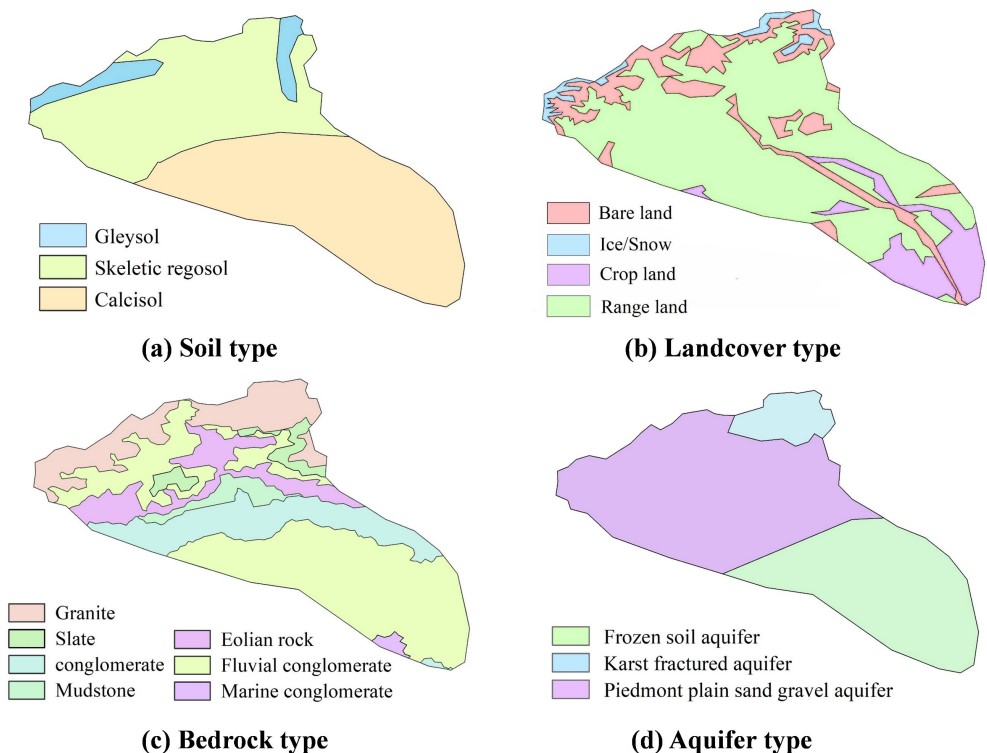

**(a) Soil type**

**(b) Landcover type**

**(c) Bedrock type**

**(d) Aquifer type**


**Figure 4.** Zonation of average tectonic uplift rates in the study area from 2000 to 2021.

As an essential driving factor in landscape evolution, the rock uplift rate data were
derived from the vertical crustal velocity field of China (Wang et al., 2020; Zubovich et al.,
2016), with the Chinese GNSS velocity field taken from Wang and Shen (2020;
https://doi.org/10.7910/DVN/C1WE3N) and the Pamir–Tien Shan GNSS velocities from
Zubovich et al. (2016; https://doi.org/10.1002/2015TC004055). The spatial distribution of
the average rock uplift rate in the study area from 2000 to 2021 was obtained through
kriging interpolation (Fig. 5).





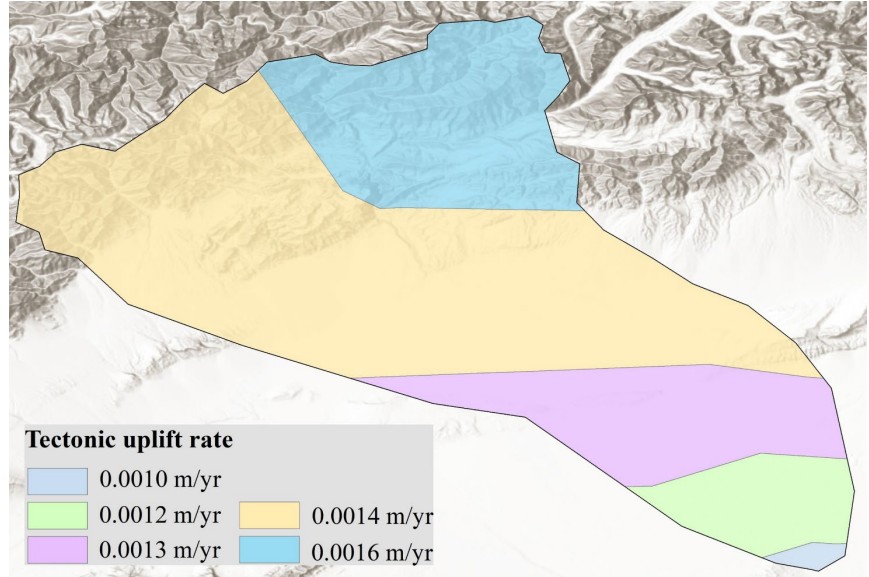

**Figure 5.** Zonation of average tectonic uplift rates in the study area from 2000 to 2021. Basemap: Esri World Hillshade (Esri).

## 3.3 Observed river channel planform data

Observed river channel planform data for December 2007, February 2014, and December 2021 were obtained using the Google Earth Pro image platform (see Fig. 6). Among these, the spatial distribution of river channels from December 2007 and February 2014 were used to identify the posterior distribution of model parameters (i.e., key parameters of LE-PIHM), while the December 2021 data were employed to validate the river channel migration results.





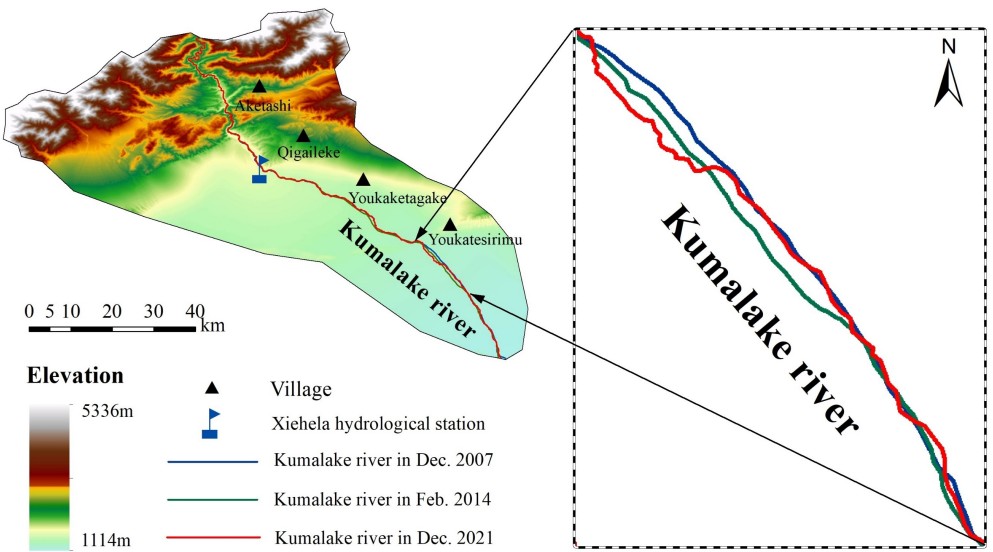

**Figure 6.** The spatial distribution of the Kumalake River channels in the study area.

## 3.4 Model settings

The outlet of the Kumalake River Basin was set at the intersection with the southeastern boundary of the study area, while all other boundaries were defined as no-flow boundaries. An unstructured triangular mesh was used to discretize the study area spatially. To accurately capture the landscape evolution processes in areas surrounding the river channel, mesh refinement was applied in the channel zones, with grid sizes less than 50 meters near the river. A total of 25,968 unstructured triangular elements were generated for the study area (see Fig. 7a).



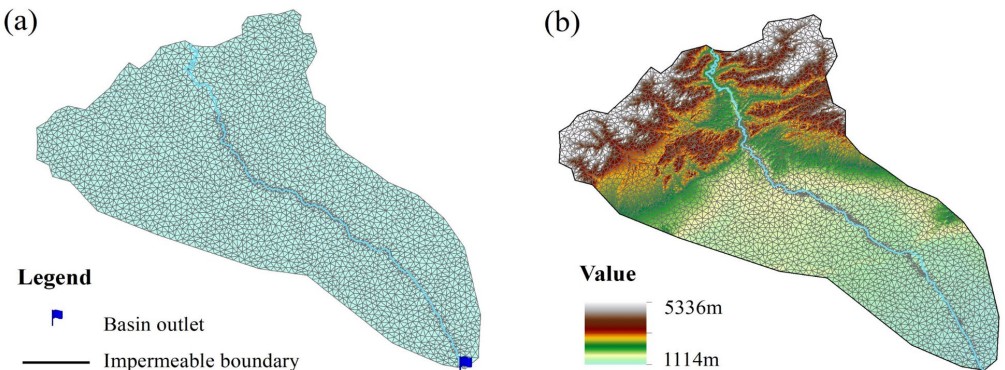

**Figure 7.** Mesh discretization and initial elevation grid of the research area.

The initial conditions required by LE-PIHM include ground surface elevation (Fig. 7b), canopy interception storage, snow depth, surface water depth, soil water storage, and groundwater storage. The canopy interception storage is initialized to 0 m in the model (Fig. 8 and Table 1).

The model was configured to run from January 2000 to December 2021, covering a total of 264 months (22 years). To capture the seasonal variability of hydrologic processes in the study area while controlling computational load, the time step was set to one month. The LE-PIHM model was executed on an Intel Xeon E5-2680 v3 server, with an average runtime per simulation of 195 minutes.





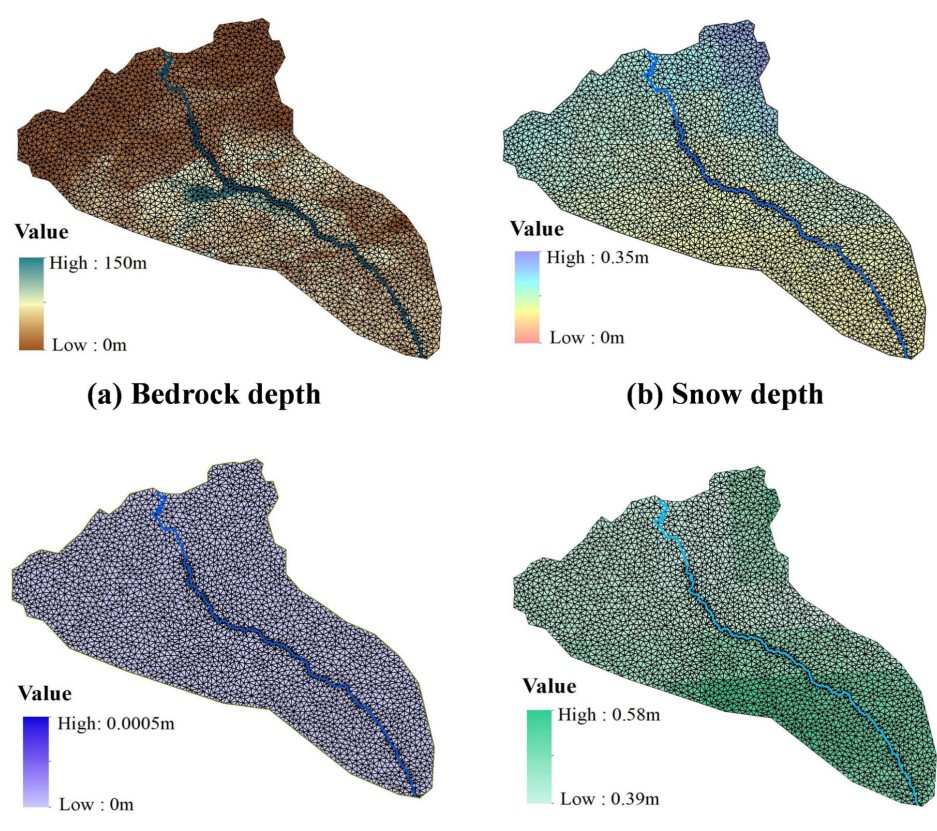

(a) Bedrock depth          (b) Snow depth

(c) Canopy interception capacity          (d) Soil water storage

**Figure 8.** The initial conditions of landscape evolution model.

## 3.5 LSTM-based surrogate model for uncertainty analysis

After constructing the basin-scale channel migration model, the next part is to conduct

parameter uncertainty analysis, followed by the reconstruction of river channel migration

over 2000–2021 and the prediction of landscape evolution and river channel changes from

2021 to 2100 under future climate scenarios.

However, performing parameter uncertainty analysis using MCMC requires a large

number of runs of the original model (i.e., the basin-scale channel migration model), which

results in a significant computational burden. To reduce this computational cost, this study



employs an LSTM network to construct a surrogate model for river channel migration. The
main steps and more details of LSTM-based surrogated models are as follows:
*i*. The Sobol method is applied for parameter sensitivity analysis. Considering the
property parameters of surface cover, soil, aquifer, and bedrock, 11 highly sensitive
parameters LE-PIHM are selected as the input variables for the surrogate model (Table 2).
**Table 2.** The parameters of landscape evolution model and their prior ranges.

| Parameters | Units | Prior distribution |
|---|---|---|
| Vegetation fractional coverage (VegFrac) | / | [0.075, 0.225] |
| Root zone depth (Rzd) | m | [0.15, 0.45] |
| Soil vertical hydraulic conductivity (KVs) | m/d | [0.30, 0.90] |
| Soil porosity (Ns) | / | [0.175, 0.525] |
| Morphological diffusivity (K1) | $m^2$/yr | [0.10, 0.30] |
| Soil particle diameter (Ds) | m | [0.0005, 0.0015] |
| Aquifer horizontal hydraulic conductivity (KHg) | m/d | [3.50, 10.50] |
| Aquifer vertical hydraulic conductivity (KVg) | m/d | [0.70, 2.10] |
| Bedrock weathering rate for bare rock (P0) | m/yr | [0.004, 0.012] |
| Tectonic uplift rate (U) | m/yr | [0.00050, 0.00155] |
| Coefficient for bedrock weathering equation (α) | 1/m | [0.01, 0.03] |

*ii*. Latin hypercube sampling is employed to generate 3,000 sample sets within the
prior ranges of the input variables.
*iii*. These 3,000 parameter sets are individually input into the LE-PIHM model to
generate surface elevation data at three time points, December 2007, February 2014, and
December 2021. Subsequently, the corresponding spatial distribution of river channel is
extracted through Arcpy scripting in ArcGIS. For each parameter set, the corresponding
river channel is discretized uniformly into 2,000 points, producing 2,000 sets of planar
coordinates of reaches.
*iv*. The obtained 3,000 sets of input variables and their corresponding coordinates are
split into training and validation datasets at 70% and 30% ratios. The LSTM network is
trained to learn the nonlinear mapping between the input variables and river channel





positions, thereby constructing a surrogate model for river channel migration. The network
terminates in a fully connected output layer with a linear activation function and He-normal
weight initialization. Training is performed using the Adam optimizer with a learning rate
of $1\times10^{-3}$, minimizing the root-mean-square error (RMSE) between the predicted and
reference river coordinate points. The model is trained for 10,000 epochs with a batch size
of 100 and a validation split of 0.1.

As shown in the validation results (Table 3), the surrogate model constructed using

LSTM demonstrates great predictive performance for river channel locations (i.e., the
planar coordinates of river reaches). It achieved a mean absolute error (MAE) of 1,411.16
m, a root mean square error (RMSE) of 1,980.10 m, and a coefficient of determination ($R^2$)
of 0.983. Furthermore, the average Hausdorff distance of the entire river between the
surrogate model and original model outputs is 229.71 m. Given the total length of river
channels in the study area is 89,337.95 m, this corresponds to only 0.25%. Thus, these
results demonstrate that the surrogate model for basin-scale river channel migration
exhibits high accuracy and reliability and can replace the original model for river channel
dynamics simulation.
**Table 3.** Evaluations of the surrogate model for basin-scale river channel migration.

| Metrics | RMSE | MSE | $R^2$ | H/Total length of river |
|---------|------|-----|-------|-------------------------|
| Results | 1411.60m | 1980.10m | 0.98 | 0.0025 |

The spatial distribution of river channels predicted by the original model and the

surrogate model are compared across three time points (Fig. 9). The surrogate model
closely matches the original in the middle and upper reaches of the river channel, while
some deviations occur in the downstream reaches. This discrepancy is primarily due to the
mountainous topography upstream, where the landscape is rugged and river channels are



relatively stable. In contrast, the downstream region is a flat plain, where river channel
migration is highly sensitive to model parameters, leading to slightly reduced accuracy in
the surrogate model's performance in that area.

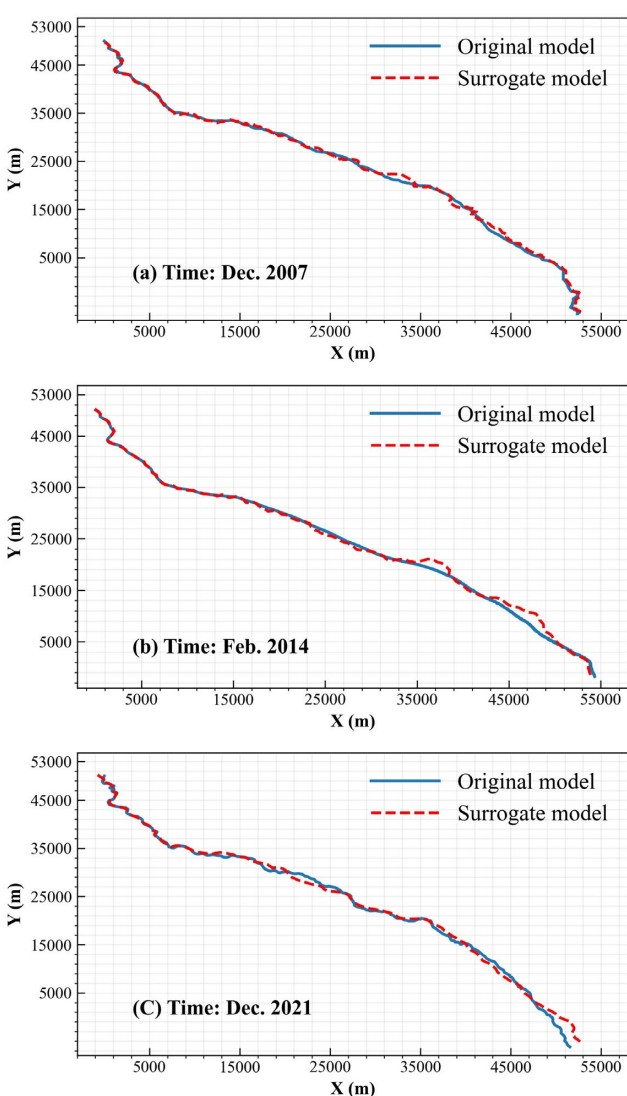


**Figure 9.** Comparison between the original river channel migration model and the
surrogate model at different time points.



Compared to the original model, the surrogate model for basin-scale river channel
migration achieves approximately a 20,000-fold increase in computational speed.
Therefore, employing this surrogate model in the parameter uncertainty analysis of river
channel migration can significantly reduce computational costs and effectively alleviate
the computational burden associated with the uncertainty analysis process.

# 396   4. Results and discussion

## 397   4.1 Parameter uncertainty analysis

### 398   4.1.1 MCMC configuration

The river channel migration model includes 11 unknown parameters to be identified,
with their prior distributions assumed to be uniform within the specified range (Table 2).
The MCMC simulation is performed using the DREAMzs sampling algorithm, with three
parallel Markov chains. Both the burn-in and formal sampling stages consist of 2,000
iterations. Additionally, the evolution of the Markov chains employs the modified
likelihood function described in Sect. 2.3.2. Based on the inferred posterior distributions
of the parameters, the posterior distribution of the river channel is obtained.

### 406   4.1.2 Posterior distributions of model parameters

The posterior probability density histograms of the river channel migration model
parameters through MCMC (Horizontal axis indicates the corresponding prior ranges; Fig.
10), together with the maximum likelihood parameter set (Table 4), show that the posterior
samples of the soil vertical hydraulic conductivity (KVs), aquifer vertical hydraulic
conductivity (KVg), soil particle diameter (Ds), bedrock weathering rate for bare rock (P0),
plant root depth (Rzd), vegetation fraction (VegFrac), and morphological diffusivity (K1)
converge to notably narrow intervals, indicating high sensitivity of these parameters to the





calibration data. In contrast, the fitting coefficient in the bedrock weathering equation (α)
and the aquifer horizontal hydraulic conductivity (KHg) exhibit broader posterior
distributions, although both display pronounced modes. Additionally, the posterior
distributions of the bedrock uplift rate (U) and soil porosity (Ns) are relatively uniform and
lack clear peak values, suggesting that considerable uncertainty remains for these
parameters after identification.

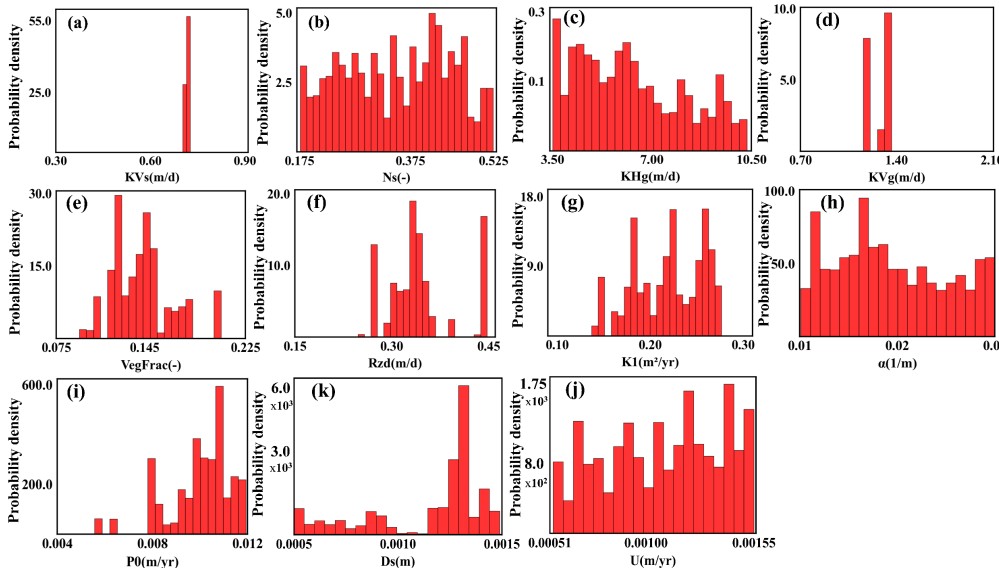


**Figure 10.** Calibrated posterior distributions of model parameters





**Table 4.** Parameter set corresponding to the maximum likelihood estimate.

| Parameters | Units | Parameter value |
|---|---|---|
| Vegetation fractional coverage (VegFrac) | / | 0.14 |
| Root zone depth (Rzd) | m | 0.17 |
| Soil vertical hydraulic conductivity (KVs) | m/d | 0.43 |
| Soil porosity (Ns) | / | 0.24 |
| Morphological diffusivity (K1) | $m^2$/yr | 0.20 |
| Soil particle diameter (Ds) | m | 0.001 |
| Aquifer horizontal hydraulic conductivity (KHg) | m/d | 9.60 |
| Aquifer vertical hydraulic conductivity (KVg) | m/d | 1.90 |
| Bedrock weathering rate for bare rock (P0) | m/yr | 0.007 |
| Tectonic uplift rate (U) | m/yr | 0.0008 |
| Coefficient for bedrock weathering equation ($\alpha$) | 1/m | 0.02 |

### 423   4.1.3 Reconstruction of river channel migration (2000-2021)

Based on the posterior distributions of the identified parameters, the river channel
evolution from 2000 to 2021 in the study area was reconstructed. The result shows that the
predicted confidence interval fully encompasses the observed river channel (Fig. 11), and
the blue shaded region denotes the 95% confidence interval, the red line represents the
observed river channel in Fig. 11. The average Hausdorff distance of the entire river
between the simulated channel (marked in black line in Fig. 11) with the maximum
likelihood parameter set and the observed channel is 225.42 m, which accounts for only
0.25% of the total river channel length. This indicates that the discrepancy between the
simulated and observed river channel per unit length is minimal, demonstrating the high
accuracy of the calibrated river channel migration model. Therefore, the river channel
model facilitated by Bayesian parameter uncertainty quantification can reliably predict the
river channel migration processes within the study area.



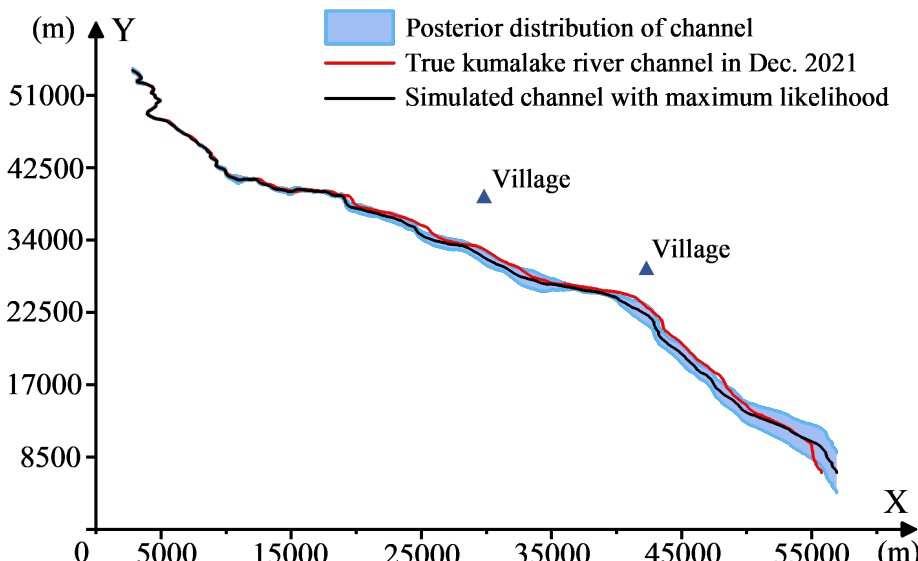


**Figure 11.** The predicted Kumalake River channel at December 2021.
Remote sensing images (Fig. 12) collected during the simulation period of the study
area indicates that, beginning in 2012, the villages along the river initiated to develop
cultivated land near the river channel. This anthropogenic activity altered key parameters
related to land cover and soil type, and may have been a driving factor for the gradual
southward migration of local river reaches. However, this mechanism of human-induced
change is not explicitly represented in the landscape evolution model, which may partly
explain the reduced simulation accuracy in the downstream region. Nevertheless, the
parameter uncertainty analysis conducted in this study helps to partially compensate for
the effects of land reclamation, keeping the prediction deviation of basin-scale river
channel migration model within an acceptable range.





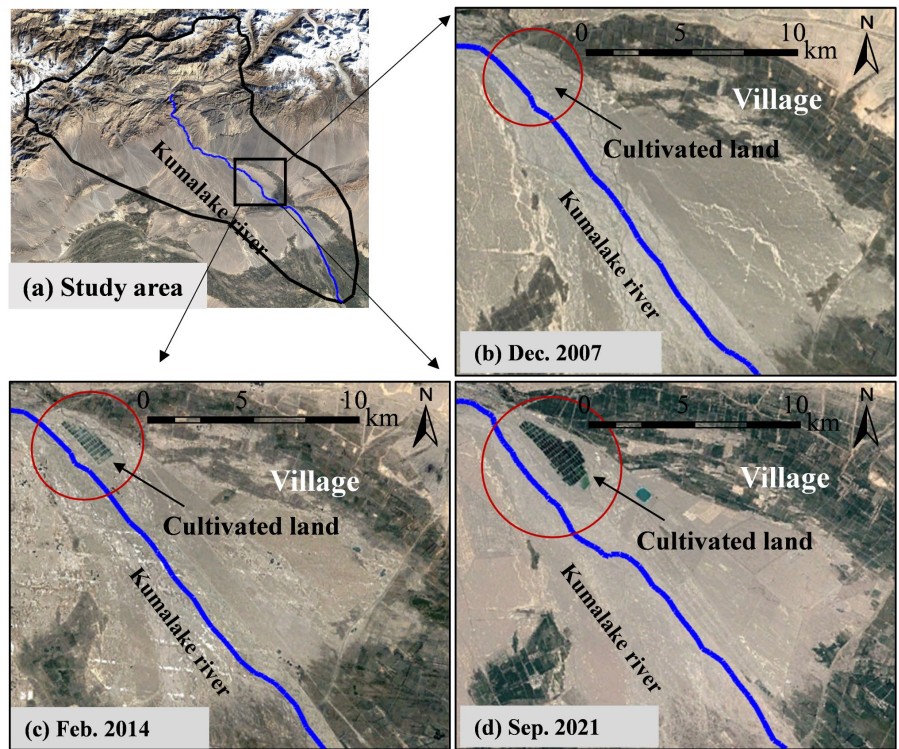

**Figure 12.** Formation process of cultivated land along the river in the study area. Satellite imagery: © Google Earth 2007, 2014, 2021, modified by the authors.

## 4.2 Prediction of river channel migration in future (2021-2100)

Future climate change will influence landscape evolution and river channel migration, thereby affecting regional water resource patterns and ecological environments. The EC-Earth3-Veg climate model, released under phase 6 of the Coupled Model Intercomparison Project (CMIP6), incorporates vegetation-climate interactions and is well-suited for evaluating terrestrial ecosystem and hydrological responses under different climate scenarios (Alsalal et al., 2024; Eyring et al., 2016). Based on the EC-Earth3-Veg model, this study adopts four scenarios of shared socioeconomic pathways (SSPs), i.e., SSP1-2.6, SSP2-4.5, SSP3-7.0, and SSP5-8.5. Each of the four climate scenarios reflects a distinct




pathway of global socioeconomic development and associated impacts on greenhouse gas
emissions and climate change (O'Neill et al., 2016). Scenario-based simulations of
landscape evolution and river channel migration were conducted by using these climate
conditions as the driving force of LE-PIHM. The climate conditions for these scenarios
over the simulation period (2021–2100) are characterized by the annual mean temperature
and precipitation (Table 5 and Fig. 13).
**Table 5**. Statistics of four climate scenarios under the EC-Earth3-Veg model.

| Climate model | Average precipitation | Average temperature | Shared Socioeconomic Pathways |
|---|---|---|---|
| | 284.97mm | 3.76°C | SSP-1.26 |
| EC-Earth3-Veg | 300.89mm | 4.05°C | SSP-2.45 |
| | 301.76mm | 4.68°C | SSP-3.70 |
| | 310.16mm | 5.33°C | SSP-5.85 |

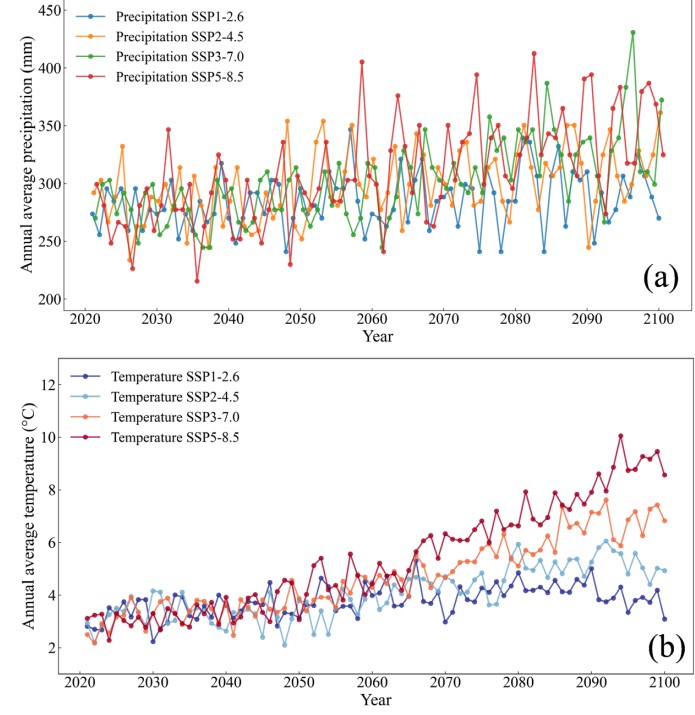


**Figure 13.** Annual mean precipitation and temperature for the four climate scenarios.





Using LE-PIHM and the parameter set corresponding to the maximum likelihood

estimate (Table 4), the landscape evolution and river channel migration from 2021 to 2100

were simulated for the study area under four climate change scenarios. After 80 years of

simulated landscape evolution, the overall topographic pattern of the region remains

characterized by higher elevations in the north and lower elevations in the south, with

elevation changes ranging from +1.39 m to −0.37 m. The simulation results under the four

EC-Earth3-Veg climate scenarios demonstrate that the elevation change and river channel

migration in basin scale by the year 2100 exhibit distinct characteristics (Fig. 14 a). In the

northern mountainous area, distinct alternating patterns of uplift and subsidence are

observed, with relatively large magnitudes of change. This is primarily due to intense

bedrock weathering and steep slopes in the mountains, where the regolith layer is thin and

the weathered soil and sediment are prone to downslope transport into adjacent low-lying

regions, resulting in significant elevation fluctuations. In contrast, the southern plain

exhibits gentle topography and minimal elevation differences, making it less conducive to

large-scale sediment transport. As a result, the landscape undulation in the plain is

relatively mild and limited in amplitude.

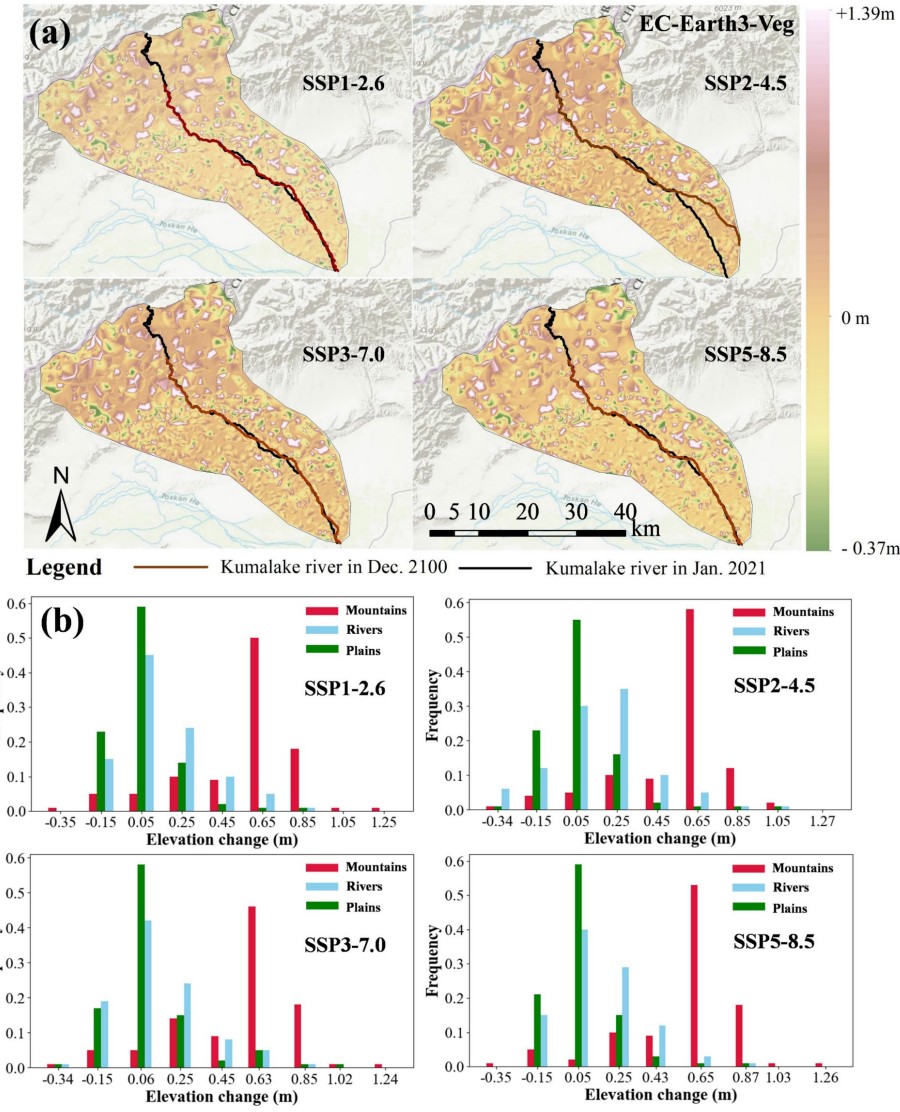

**Figure 14.** Spatial distribution of river channel (a) and Elevation change (b) under four

future climate scenarios based on the EC-Earth3-Veg climate model. Basemap: Esri

World Hillshade (Esri).

Under the EC-Earth3-Veg climate model, the four future climate scenarios induce

distinct patterns of river channel migration. By the end of the simulation period (2100), the

upstream reaches of the Kumalake River remain largely unchanged from their 2021





positions. This stability is attributed to the fact that these reaches flow through narrow and
geomorphologically stable canyon landscape, where external disturbances are minimal and
the river alignment and morphology remain relatively constant. In contrast, several
downstream reaches situated in the plain exhibit varying degrees of northward migration.
This phenomenon is mainly driven by the relatively gentle topography of the plains,
increased sediment deposition, redistribution of flow energy, and anthropogenic activities.
These factors collectively lead to channel swinging, incision, or aggradation, gradually
shifting the river channel position.

The four scenarios under EC-Earth3-Veg exhibit significant differences in

temperature and precipitation characteristics, as well as in the extent of river channel
migration (Table 6). Notably, under the SSP2-4.5 scenario, although the climate variability
is not the most intense among the scenarios, the elevation change shows the largest
amplitude, and the topographic changes in the river area are more pronounced (Fig. 14 b),
leading to a shift of the downstream river channels into the plain area. The specific
combination of climatic conditions may cause the river system to approach an evolutionary
threshold (Church et al., 2002; Meyer et al., 2018). When such a critical threshold is
reached, the river system may undergo abrupt transitions, manifesting as regime shifts
(Church et al., 2002). Under this scenario (SSP2-4.5), the downstream river channel
deviates entirely from its initial position, with an average Hausdorff distance reaching
1,873.37 m, primarily in the northward direction. This suggests that the climate changes in
this scenario trigger a sudden and intense spatial reorganization of the river network,
resulting in significant morphological transformation.

These results indicate that the process of river channel migration in the study area



under climate forcing varies across scenarios. Future climate conditions play a critical role
in regional landscape evolution and river dynamics. Furthermore, the complex feedbacks
between climate change and geomorphic systems highlight the importance of incorporating
these interactions in predictive modeling of fluvial landscape evolution and watershed
management planning.
**Table 6.** Statistics of climate scenarios and average Hausdorff distance of river migration.

|  | SSP1-2.6 | SSP2-4.5 | SSP3-7.0 | SSP5-8.5 |
|---|---|---|---|---|
| Variance of precipitation ($mm^2$) | 576 | 922.45 | 1227.62 | 1942.72 |
| Range of precipitation (mm) | 105.85 | 127.75 | 186.15 | 197.1 |
| Variance of temperature (℃$^2$) | 0.34 | 0.92 | 1.99 | 4.27 |
| Range of temperature (℃) | 3.08 | 3.96 | 5.45 | 7.79 |
| H(m) | 333.26 | 1873.37 | 405.69 | 304.85 |

## 5. Conclusions

River channel migration at the basin scale not only determines the spatial distribution
pattern of regional river networks, but also exerts profound influences on local ecosystems
and the development of civilizations within the basin. Simulating river channel migration
at the basin scale aids in quantitatively reconstructing this long-term, complex dynamic
processes and also provides a scientific basis for decision-making in response to climate
change and natural disasters. To address the limitations of traditional river channel
migration models in temporal and spatial scale applications, this study integrates a
landscape evolution model with river channel extraction techniques, achieving accurate
and reliable simulation of river channel migration processes at the basin scale. Using the
Kumalake River Basin as a case study, the river channel migration process in the region is
reconstructed based on the LE-PIHM landscape evolution model and river channel
extraction techniques. The main conclusions of this study are as follows:



1. The LSTM-based surrogate model for river channel migration demonstrates high
accuracy and effectively overcomes the computational challenge associated with parameter
uncertainty analysis. The parameter calibration using MCMC requires numerous
executions of the LE-PIHM model and river channel extraction, resulting in prohibitive
computational demands. The surrogate model for basin-scale river channel migration based
on LSTM networks accurately characterizes the response relationship between landscape
evolution parameters and river channel locations, effectively solving the problem of
computational burden in Bayesian uncertainty analysis.
2. The river channel migration model facilitated by Bayesian parameter uncertainty
quantification can reliably predict the river channel evolution process within the study area.
Based on the inferred posterior distributions of model parameters, the predicted confidence
interval of the channel fully encompasses the actual river location. The average Hausdorff
distance between the simulated river channel with the maximum likelihood parameter set
and the observed river channel is 225.42 m, which accounts for only 0.25% of the total
channel length. Thus, the basin-scale river channel migration model incorporating
Bayesian uncertainty analysis demonstrates high reliability and predictive capability,
enabling effective characterization of river migration processes within the study area.
3. River channel evolution under different climate scenarios demonstrates significant
variability, and future climate change will profoundly affect basin geomorphological
characteristics and river network configurations. Based on the EC-Earth3-Veg model
released by CMIP6, the landscape evolution and river channel migration in the study area
from 2021 to 2100 were projected under four Shared Socioeconomic Pathways (SSP1-2.6,



SSP2-4.5, SSP3-7.0, and SSP5-8.5). The results indicate that climate change and
geomorphological systems exhibit complex response mechanisms.
***Author contributions.***
JW and XZ conceptualized the study and designed the research methodology. JW and
QW conducted the simulations and implemented the methodology. JW produced all the
figures and tables. XZ and JW contributed to the data validation and data curation. DW
supervised the research. All authors reviewed, edited and approved the final version of the
manuscript.
***Competing interest.***
The contact author has declared that none of the authors has any competing interests.
***Data Availability.***
The temperature and precipitation data used in this study from the World Data Center
for Climate (WDCC) are open-access and publicly available: EC-Earth-Consortium EC-
Earth3-Veg (https://doi.org/10.26050/WDCC/AR6.C6CMEEEVE, CMIP6). The observed
river channel planform data used for uncertainty analysis mentioned in Sect. 3.3 have been
made        publicly        available        via        the        Hydroshare        platform
(https://doi.org/10.4211/hs.a6eb2a2c8ae746cf99d5d89a5ed2600b, Zeng and Wu., 2025).
***Acknowledgments.***
This study was supported by the National Key Research and Development Program
of China (2024YFC3713001), the National Natural Science Foundation of China
(42477082). We are grateful to the High-Performance Computing Center (HPCC) of
Nanjing University for performing the simulations in this paper.




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
