# Peer review of "Prediction of basin-scale river channel migration based on"

_EGUsphere, 2025_

## Author Comment (AC1)

**RC1**

**Prediction of basin-scale river channel migration based on landscape evolution numerical simulation**

Jitian Wu, Xiankui Zeng, Qihui Wu, Dong Wang, and Jichun Wu

We sincerely thank the reviewer for their insightful and constructive comments. We have carefully addressed each point below and will incorporate the corresponding revisions into the manuscript. We hope that the revised manuscript has met the quality standards for publication in HESS.

Note:

(1) In this response, the text in *italic type* is the original comments from the reviewers, and the text in blue, headed with "Reply", is the response from the authors.

(2) In the manuscript, the words in blue indicate the sentence is improved or revised. Some of them are mentioned in this response via the page and line number.

**Response to Reviewers**

**Comments:**

(1) *The parameter uncertainty is performed by using Markov Chain Monte Carlo method, and a modified Gaussian likelihood function is used. It is interesting in Bayesian uncertainty analysis. However, the statistical assumptions behind Equation (10) are still somewhat unclear, and further explanation is recommended. What is the physical meaning of $\Sigma$ in Equation (10)? Could non-Gaussian likelihood functions or different error model specifications further improve results?*

Reply: Reply: Thank you to the reviewer for pointing out these two issues.

① In this study, Eq. (10) is formulated as a modification of the classical Gaussian likelihood in Eq. (6). The key assumption is that the average Hausdorff distance (H) between the simulated and observed channel planforms follows a zero-mean, independent and identically distributed Gaussian error model, i.e., $H \sim \mathrm{N}(0, \sigma^2)$. Under

this setting, $H_{obs}$ represents the distance between the true channel and itself (thus $H_{obs}= 0$), whereas H denotes the distance between the model-simulated channel planform and the true channel.

$$L(\theta^i \mid D) = \frac{1}{2\pi^{n/2} |\Sigma|^{1/2}} \exp\left[ -\frac{[D - f(\theta^i)]^T \Sigma^{-1} [D - f(\theta^i)]}{2} \right] \qquad (6)$$

$$\ln L = -\frac{1}{2}[\frac{H^2}{\Sigma^2} + \ln(2\pi\Sigma^2)] \qquad (10)$$

In Eq. (6), $\Sigma$ denotes the covariance matrix of the residual errors. Meanwhile,

$$D - f(\theta^i) = H - H_{obs} = H - 0 = H$$

Therefore, in Eq. (10), $\Sigma$ is no longer the covariance matrix used in the conventional Gaussian likelihood; instead, it represents the variance of $H$.

$$\ln L = -\frac{1}{2}\left[ \frac{H^2}{\sigma^2} + \ln(2\pi\sigma^2) \right]$$

In addition, we have revised Eq. (10) accordingly in the revised manuscript.

② We sincerely appreciate the reviewer's insightful comments regarding the choice of likelihood function. We agree that incorporating non-Gaussian likelihoods or more complex residual structures could potentially enhance predictive performance, such as the correlated, heteroscedastic, and non-Gaussian functions. However, adopting a non-Gaussian likelihood generally entails additional hyperparameters, which increases parameter uncertainty and renders the convergence of MCMC algorithms more challenging, thereby substantially raising computational cost. Given the limited amount of data available for parameter estimation and the absence of clear evidence of heteroscedasticity in the residual diagnostics, the Gaussian likelihood provides a reasonable balance between model reliability and computational feasibility.

(2) *The manuscript mentioned that since 2012, there has been significant agricultural development in the downstream river reaches, and human activities may have altered land cover, soil properties, and river channel constraints. However, in the model, the*

*settings for land cover and soil parameters do not seem to be influenced by time. This is an important limitation and should be more clearly emphasized. Currently, it is briefly mentioned only as a qualitative explanation for local mismatches.*

Reply: Thank you to the reviewer for providing this important suggestion. We agree that the rapid intensification of agricultural activities in the downstream reach since 2012 may have influenced channel evolution by altering land-cover types and associated soil physical properties. However, in this study, the relevant landscape-evolution parameters were not parameterized as time-varying, for the following reasons.

First, although remote-sensing imagery allows a qualitative identification of cropland expansion in the downstream area after 2012, continuous and reliable land-cover and soil-property datasets at the study-basin scale are not available. Under such data limitations, introducing time-varying landscape-evolution parameters would likely increase parameter uncertainty and consequently reduce the robustness of the model predictions.

Second, while the landscape-evolution model does not explicitly account for temporal changes in parameters driven by human activities, we conducted a Bayesian parameter uncertainty analysis that effectively enables key model parameters to adjust adaptively within physically plausible ranges through parameter identification. By comparing the simulated and observed channel planform distributions, we verified that the identified parameter set can reasonably represent the dominant landscape-evolution characteristics of the study area.

(3) *In section 3.2, the datasets from NASA (Leaf Area Index, Surface Roughness, Air Temperature) are referenced. The resolution of these input raster datasets is relatively coarse. Could this impact the accuracy of the simulations?*

Reply: Thanks for this comment. The objective of this study is to simulate basin-scale, long-term channel migration, rather than short-term, reach-scale hydrodynamic evolution. At these spatial and temporal scales, the response of LE-PIHM to meteorological and hydrological forcing is primarily reflected in the basin-wide water balance and the cumulative effects of long-term erosion–deposition, rather than in a

detailed representation of high-frequency processes and fine-scale spatial heterogeneity (Tucker and Hancock, 2010; Coulthard and Skinner, 2016; Zhang et al., 2016). Therefore, using relatively coarse-resolution datasets for basin-scale landscape-evolution simulations, together with a parameter-uncertainty analysis, can achieve an acceptable level of accuracy. Moreover, such datasets have been widely adopted in large-scale watershed hydrologic modeling studies (Asong et al., 2020; Gelaro et al., 2017; Qi et al., 2015; Rodell et al., 2004).

(4) *The simulation technique for basin-scale river channels proposed in the manuscript has been successfully applied to the Kumalake River Basin. A broader discussion of the generalizability of this method would help improve its applicability.*

Reply: Thank you to the reviewer for this important suggestion. We agree that it is necessary to discuss the generalizability of the proposed basin-scale channel-migration modeling framework, and we have added the corresponding discussion in the revised manuscript. Specifically, we address the generalizability of the approach from the following perspectives:

(1) Generality of the modeling framework. The proposed technique is built upon LE-PIHM and a DEM-driven channel-extraction procedure. Because it does not rely on basin-specific assumptions or a particular basin type, it can be transferred to other basins provided that basic topographic, climatic, and geological datasets are available.

(2) Adaptability to different dominant controls. The framework explicitly couples hydrological processes, landscape evolution, and tectonic uplift. Key parameters (e.g., erodibility coefficients, uplift rates, and hydrologic parameters) can be adjusted according to local conditions, enabling application to basins primarily controlled by climate forcing or by tectonic activity.

(3) Transferability of the uncertainty analysis and surrogate modeling components. The LSTM surrogate model and the Hausdorff distance-based modified likelihood are not basin-specific. Their training workflow and the associated parameter-uncertainty analysis framework can be directly transferred to other basin-scale channel-migration studies, indicating strong methodological portability.

Meanwhile, we also clarify in the manuscript the potential limitations of the approach in basins that are strongly regulated by human engineering interventions (e.g., channelization projects or large-reservoir operations). We further discuss possible future extensions, such as introducing parameterizations of human activities, to broaden its applicability. We believe these additions more comprehensively demonstrate the transfer potential and application prospects of the proposed framework across diverse basin settings.

(5) *A marked disparity in the extent of river channel migration is evident between the upstream and downstream reaches of the basin (Figure 11). The mechanisms underlying this phenomenon require further explanation.*

Reply: We thank the reviewer for pointing out this issue. In our study basin, the upstream area is mountainous, whereas the downstream area is a lowland plain. The relatively narrow migration envelope (i.e., lower uncertainty) in the upstream reach is primarily because this segment is confined within a canyon setting where the valley is topographically narrow and comparatively stable. As a result, the channel is strongly constrained laterally, and the channel-migration model exhibits lower predictive uncertainty in this reach, leading to a narrower simulated distribution of channel positions.

In contrast, parts of the downstream reach located on the plain show substantially larger predictive uncertainty. This is because the low-relief terrain provides fewer topographic constraints on lateral migration, and channel behavior is more strongly influenced by the combined effects of deposition, changes in flow hydraulics, and human activities. Consequently, the basin-scale river channel migration model is associated with greater uncertainty in the downstream plain, yielding a wider predicted channel distribution, indicating higher predictive uncertainty.

(6) *In the future scenario of SSP2-4.5 (Figure 14), significant river channel reorganization occurs, and the elevation changes in the river segments under this scenario are also noticeable, which is very interesting. What are the underlying mechanisms causing this phenomenon?*

Reply: Thanks for this insightful comment. We argue that the pronounced channel migration and the large reach-scale elevation changes under SSP2-4.5 are not solely driven by the magnitude of climate change itself, but rather reflect a nonlinear, threshold-like response of the fully coupled hydro–geomorphic system under specific climatic forcing.

Within the LE-PIHM landscape evolution framework, channel planform dynamics and elevation changes are jointly controlled by the following processes:

(1) Precipitation–runoff mechanisms, including rainfall, infiltration and runoff generation, and surface–groundwater exchange;

(2) Sediment supply and transport capacity, including hillslope diffusion, weathering-driven sediment production, and river sediment transport;

(3) Landscape–flow-routing feedbacks, whereby landscape evolution and the associated adjustment of D8-based flow paths modify local slope and discharge concentration.

Under the SSP2-4.5 scenario, the combined effects of these mechanisms drive downstream lowland reaches toward a critical geomorphic threshold, thereby triggering obvious channel migration.

(7) *It would be beneficial to add information on the variability (such as the standard deviation) of precipitation and temperature across the different scenarios in Table 5.*

Reply: We have added the variances and ranges of precipitation and temperature in Table 6 of the revised manuscript. The results show that the different scenarios exhibit not only substantial differences in the mean values of precipitation and temperature, but also clear scenario-dependent variability. In general, higher-emission scenarios are associated with larger fluctuations, reflected by greater variances and wider ranges in both temperature and precipitation.

(8) *To help readers distinguish the variables for the four climate scenarios, the line colours in Figure 13 should be redesigned.*

Reply: Thanks for pointing out this issue. As shown in Fig. 13, we have adjusted the

[Figure]

line colors accordingly in the revised manuscript.

**Figure 13.** Annual mean precipitation and temperature for the four climate scenarios.

**Reference:**

Asong, Z. E., Elshamy, M. E., Princz, D., et al. High-resolution meteorological forcing data for hydrological modelling and climate change impact analysis in the Mackenzie River Basin, Earth Syst. Sci. Data, 12, 629–645, 2020.

Gelaro, R., McCarty, W., Suárez, et al. The Modern-Era Retrospective Analysis for Research and Applications, Version 2 (MERRA-2), J. Climate, 30, 5419–5454, 2017.

Qi, W., Zhang, C., Fu, G., et al. Global Land Data Assimilation System data assessment using a distributed biosphere hydrological model, J. Hydrol., 528, 652–667, 2015.

Rodell, M., Houser, P. R., Jambor, U., et al. The Global Land Data Assimilation System, Bull. Amer. Meteor. Soc., 85, 381–394, 2004.

Schoups, G. and Vrugt, J. A.: A formal likelihood function for parameter and predictive inference of hydrologic models with correlated, heteroscedastic, and non-Gaussian

errors, Water Resour. Res., 46, W10531, https://doi.org/10.1029/2009WR008933, 2010.

Vrugt, J. A., de Oliveira, D. Y., Schoups, G., and Diks, C. G. H.: On the use of distribution-adaptive likelihood functions: Generalized and universal likelihood functions, scoring rules and multi-criteria ranking, J. Hydrol., 615, 128542, https://doi.org/10.1016/j.jhydrol.2022.128542, 2022.

Evin, G., Kavetski, D., Thyer, M., and Kuczera, G.: Pitfalls and improvements in the joint inference of heteroscedasticity and autocorrelation in hydrologic model calibration, Water Resour. Res., 49, 4518–4524, https://doi.org/10.1002/wrcr.20284, 2013.

Choi, S. Y., Seo, I. W., and Kim, Y.-O.: Parameter uncertainty estimation of transient storage model using Bayesian inference with formal likelihood based on breakthrough curve segmentation, Environ. Model. Softw., 123, 104558, https://doi.org/10.1016/j.envsoft.2019.104558, 2020.

Tucker, G. E. and Hancock, G. R.: Modelling landscape evolution, Earth Surf. Process. Landf., 35, 28–50, https://doi.org/10.1002/esp.1952, 2010.

Coulthard, T. J. and Skinner, C. J.: The sensitivity of landscape evolution models to spatial and temporal rainfall resolution, Earth Surf. Dynam., 4, 757–771, https://doi.org/10.5194/esurf-4-757-2016, 2016.

Zhang, Y., Slingerland, R., and Duffy, C.: Fully-coupled hydrologic processes for modeling landscape evolution, Environ. Model. Softw., 82, 89–107, https://doi.org/10.1016/j.envsoft.2016.04.014, 2016.

---

## Author Comment (AC2)

**RC2**

**Prediction of basin-scale river channel migration based on landscape evolution numerical simulation**

Jitian Wu, Xiankui Zeng, Qihui Wu, Dong Wang, and Jichun Wu

We sincerely thank the reviewer for their thoughtful and constructive feedback. We have carefully considered each comment, provided point-by-point responses below, and revised the manuscript accordingly. We believe these revisions have strengthened the study, and we hope the updated manuscript now meets the standards for publication in *Hydrology and Earth System Sciences* (HESS).

Note:

(1) In this response, the text in *italic type* is the original comments from the reviewers, and the text in blue, headed with "Reply", is the response from the authors.

(2) In the manuscript, the words in blue indicate the sentence is improved or revised. Some of them are mentioned in this response via the page and line number.

**Comments:**

1. *The manuscript should explicitly clarify the statistical/error-model assumptions behind using the average Hausdorff distance H as the calibration target and then use the same metric to provide a reach-wise (regional) difficulty diagnosis for the channel prediction/surrogate. Currently, the likelihood formulation based on H (with $H_{obs} = 0$) is presented, but the assumed distribution for H and the selection/estimation of the scale (variance) term are not sufficiently specified, which directly affects posterior tightness and the credibility of uncertainty bounds. Also, the paper should quantify the known spatial heterogeneity in performance by splitting the main channel into two or three reaches (e.g., upstream canyon vs downstream plain) and reporting H (and optionally mean pointwise distance) per reach for both reconstruction/validation and surrogate evaluation, since the text indicates downstream deviations are larger. This will make the Bayesian calibration statistically transparent while also giving readers a practical, spatially explicit statement of where the workflow is reliable.*

Reply: Thank you to the reviewer for pointing out these two issues.

① In this study, the average Hausdorff distance (H) is used to quantify the spatial discrepancy between the simulated and observed channel centerlines (i.e., two curves). The observed value $H_{obs}$ represents the distance between the true channel and itself (thus $H_{obs} = 0$), whereas the simulated value H represents the distance between the modeled channel output and the true channel. In the Bayesian uncertainty analysis, we assume that the error in H follows a zero-mean, independent and identically distributed Gaussian distribution, and we construct the likelihood function accordingly. We have clarified this assumption in the revised manuscript: the mean Hausdorff distance H is treated as an average measure of the spatial deviation between the simulated and observed channel planforms, and its observation error is assumed to be normally distributed, i.e., $H \sim N(0, \sigma^2)$.

② We agree with the suggestion to evaluate spatial heterogeneity in model performance by assessing metrics along different segments of the main channel. Dividing the main channel by geomorphic units—upstream canyon reach, midstream transitional reach, and downstream alluvial-plain reach—and computing the average Hausdorff distance (H) for each segment would explicitly account for spatial heterogeneity and could provide a more informative diagnosis of predictive skill. However, if segmented reaches are used within the MCMC-based parameter uncertainty analysis, the variance/covariance specification in the likelihood (Eq. 10) must be defined appropriately for each reach. Because geomorphic setting, channel stability, and migration amplitude differ substantially among reaches, the variance (or covariance-matrix elements) should be parameterized in a reach-dependent manner. An inappropriate specification would bias the likelihood evaluation and compromise the accuracy of the inferred posterior distributions and uncertainty bounds. For these reasons, we performed parameter calibration and uncertainty quantification using a basin-wide (whole-channel) computation of H. The results indicate that this whole-channel approach can effectively characterize the channel-migration process over the study period, and the resulting error level is adequate for the objectives of this study.

In future work, we will consider explicitly partitioning the main channel into three reaches (upstream canyon, midstream transition, and downstream alluvial plain) and further investigating how to specify the variance term (or covariance structure) in Eq. (10) under a segmented formulation, with the goal of enabling a finer and more robust characterization of reach-scale uncertainty and spatial heterogeneity.

2. *The LSTM surrogate section should be expanded with minimal but essential implementation details to ensure reproducibility, beyond the already provided training design (LHS sample size, train/validation split, optimizer and hyperparameters). Specifically, please add a compact description (ideally a short table plus a few sentences) of the LSTM architecture (number of layers, hidden units, dropout/regularization if any), the preprocessing applied to the 11 parameters (e.g., min–max scaling or z-score normalization), the exact output formatting of the 2,000-point planform, and the loss definition used to train the network (e.g., coordinate-wise MSE, any weighting along the channel). These additions are documentation-level and do not require new experiments, but they materially improve the scientific value of the surrogate contribution by allowing other groups to replicate and benchmark the approach.*

Reply: Thank you to the reviewer for providing this important suggestion. We have added additional implementation details for constructing the LSTM-based surrogate model in the revised manuscript. The surrogate model employs a two-layer LSTM architecture followed by a linear fully connected layer, taking normalized LE-PIHM parameters as inputs and outputting planar river-channel coordinates. The network is trained with the Adam optimizer by minimizing the RMSE between the predicted and reference channel planforms. Table 2 summarizes the specific LSTM configuration and training settings as follows:

**Table 2.** Architecture of the LSTM surrogate model

| Training design | Specification |
| --- | --- |
| Input parameters | 11 key LE-PIHM parameters |
| Input preprocessing | Min-max normalization applied to each parameter based on its prior range, scaled to [0, 1] |
| Network architecture | Two stacked LSTM layers followed by one fully connected a dense layer |
| LHS sample size | 3000 parameter sets |
| Training set size | 2100 samples (70% of LHS sample size) |
| Validation set size | 900 samples (30% of LHS sample size) |
| LSTM layer 1 | 128 hidden units; return full output sequence |
| LSTM layer 2 | 256 hidden units; return last time-step output only |
| Model output | Planform locations of river channel represented by 2000 uniformly sampled points |
| Output format | (x, y) coordinates of channel points |
| Loss function | RMSE between surrogate predicted and reference channel coordinates |
| Optimizer | Adam |
| Learning rate | 0.001 |
| Batch size | 100 |
| Training epochs | 10,000 |

3. *The future-scenario projection component should include a clear, concise description of how CMIP6 EC-Earth3-Veg forcings under the SSP scenarios are prepared and mapped into LE-PIHM, because scenario-to-scenario differences in projected migration—particularly the large response reported for SSP2-4.5—are sensitive to bias correction, downscaling, and temporal aggregation choices. Please state explicitly whether precipitation/temperature are used raw or bias-corrected (and name the method at a high level if applied), how spatial downscaling/interpolation to the basin model grid is performed, and what temporal resolution is used to drive the model during 2021–2100 (including whether any aggregation is performed before the landscape-evolution time step). A brief paragraph should then connect the interpretation of "threshold-like" migration behavior to these forcing-preprocessing uncertainties; this strengthens the credibility of the scenario comparison.*

Reply: Thank you for pointing out that the description of CMIP6 forcing preparation and processing for the future-scenario simulations was insufficient in the original

manuscript. We agree that scenario-to-scenario differences in projected channel migration may be sensitive to forcing-preprocessing choices, including bias correction, spatial downscaling/interpolation, and temporal resolution.

Accordingly, we have added a clarifying paragraph in Section 4.2 of the revised manuscript: "In this study, future-scenario projections are driven by CMIP6 EC-Earth3-Veg precipitation and air-temperature data under SSP1-2.6, SSP2-4.5, SSP3-7.0, and SSP5-8.5. The scenario forcing fields are mapped to the LE-PIHM basin computational units using bilinear interpolation. To maintain consistency with the monthly time step adopted in LE-PIHM, the forcing data are temporally aggregated to monthly resolution prior to being used as model inputs. In addition, no bias correction is applied to the CMIP6 forcing data in this study."

---

## Author Comment (AC3)

**RC3**

**Prediction of basin-scale river channel migration based on landscape evolution numerical simulation**

Jitian Wu, Xiankui Zeng, Qihui Wu, Dong Wang, and Jichun Wu

We are grateful to the reviewer for the insightful and constructive comments. We have carefully considered each point, provided a detailed point-by-point response below, and revised the manuscript accordingly. We believe these changes have improved the clarity and scientific rigor of the work, and we hope the revised manuscript now meets the publication standards of *Hydrology and Earth System Sciences* (HESS).

Note:

(1) In this response, the text in *italic type* is the original comments from the reviewers, and the text in blue, headed with "Reply", is the response from the authors.

(2) In the manuscript, the words in blue indicate the sentence is improved or revised. Some of them are mentioned in this response via the page and line number.

**Response to Reviewers**

**Comments:**

*1. This study uses the average Hausdorff distance as an indicator to assess the difference between the simulated and the observed river channel, thereby evaluating the simulation accuracy. What is the justification for this indicator?*

Reply: Thank you for raising this important question regarding the choice of evaluation metric. In this study, we use the average Hausdorff distance (H) to quantify the discrepancy between the simulated and observed channels, and its suitability is supported by the following considerations.

First, our focus is on basin-scale, long-term channel migration, rather than short-term hydrodynamics or reach-scale morphological details. The average Hausdorff

distance (H) provides a direct measure of the planform geometric offset between two channel centerlines (see figure below). As a distance-based metric with clear physical meaning, it effectively characterizes the overall positional deviation of the simulated channel relative to the true channel, and is therefore well suited to our modeling objective.

In addition, the Hausdorff distance is a well-established measure of curve-to-curve spatial similarity, and has been widely used in studies of channel-shape matching, path comparison, and geomorphic feature analysis (Lei and Lei, 2022; Bogoya et al., 2019; Ranacher and Tzavella, 2014). Compared with metrics that evaluate discrepancies at individual points or local locations, the average Hausdorff distance (H) summarizes distances across all discretized points along the entire channel, thereby accounting for both the overall displacement and differences in channel curvature and planform geometry.

[Figure]

**Figure R1.** Schematic diagram of Hausdorff distance.

2. *The results of the parameter uncertainty analysis in Figure 10 show the posterior distribution histograms of model parameters, which are very useful. The significance of these posterior distributions could be further explained. I recommend to provide*

*additional discussion on the posterior ranges.*

Reply: We thank the reviewer for pointing out this issue. A more in-depth discussion of the posterior parameter distributions in Section 4.1.2 can help readers better understand model behavior and the mechanisms controlling channel migration in the basin scale.

The posterior distributions in Fig. 10 indicate that the inferred posteriors not only quantify uncertainty in parameter identification, but also reveal the relative importance of different geomorphic and hydrologic processes for reproducing channel migration in the basin scale. Specifically, several parameters—such as the vertical soil hydraulic conductivity (KVs), hillslope diffusion coefficient (K1), bare-bedrock weathering rate (P0), soil grain size (Ds), and vegetation fraction (VegFrac)—exhibit clear convergence to relatively narrow and concentrated posterior ranges after calibration. This suggests that channel-planform predictions are highly sensitive to these parameters, and that they exert strong control on basin-scale erosion–deposition balance and channel migration behavior. From a geomorphological perspective, these parameters directly regulate hillslope sediment supply, runoff generation, and fluvial sediment-transport capacity, and therefore constitute key controls on the magnitude of channel-position adjustment.

In contrast, the posterior distributions of some parameters—such as aquifer horizontal hydraulic conductivity (KHg), the weathering-law coefficient (α), and the tectonic uplift rate (U)—remain comparatively broad. This indicates that, given the spatial scale of this study and the available observational constraints, channel-planform position is less responsive to these parameters. Over the past 22 years, channel migration in the basin scale may have limited sensitivity to tectonic uplift, or the effects of uplift may be partially compensated by other parameters within the coupled hydro–geomorphic system.

[Figure]

**Figure 10.** Calibrated posterior distributions of model parameters

3. *River channel migration at the basin scale is the result of landscape evolution processes. The manuscript proposes that LE-PIHM includes processes such as tectonics and hydrology. Could the authors provide more detailed explanations of the full-coupled multi-processes involved?*

Reply: Thank you to the reviewer for providing this important suggestion. LE-PIHM is a fully coupled watershed landscape-evolution model that integrates surface–subsurface hydrologic processes, snow accumulation and melt, hillslope and channel sediment transport, weathering and erosion, and tectonic uplift.

The hydrologic and geomorphic modules are tightly coupled within the same control volume through mass conservation and flux closure, and the state variables are updated synchronously at each time step. For each grid element (a Triangulated Irregular Network, TIN, control volume), the model simultaneously tracks seven state variables: canopy water storage, snowpack, surface-water depth, vadose-zone water storage, saturated-zone groundwater table, land-surface elevation (z), and bedrock-interface elevation (e). These state variables are assembled into a unified global system of ordinary differential equations (ODEs) and are solved concurrently. Within each TIN control volume, multiple fluxes—including infiltration, recharge, overland flow,

groundwater flow, and sediment/weathering/uplift fluxes—coexist and are transported in a coupled manner across the entire TIN mesh domain.

Specifically, the hydrologic module represents the sequence of processes from precipitation to canopy interception/snowmelt, surface runoff generation and routing, infiltration, recharge to the vadose and saturated zones, lateral groundwater flow, and evapotranspiration. Overland flow is parameterized using the Manning equation, and lateral groundwater flow is represented using Darcy's law, with fluxes controlled by hydraulic-head gradients and unsaturated hydraulic conductivity. The geomorphic module updates land-surface elevation z and bedrock-interface elevation e based on mass conservation, with key controls including the tectonic uplift rate (U), bedrock weathering rate, and hydraulically driven sediment fluxes on hillslopes.

Topography elevation and bedrock elevation determine hydraulic slopes and head gradients, whereas hydrologic states (e.g., surface-water depth and groundwater level) govern shear stress and sediment-transport capacity. Sediment fluxes (qs), hillslope fluxes (qc), and uplift terms are solved in a coupled manner to update elevations, which in turn modify hydraulic gradients and flow-routing patterns. This two-way feedback implies that the evolving landscape constrains river dynamics, and river processes reshape the landscape, thereby forming a self-consistent loop of hydro–geomorphic co-evolution.

4. *For the parameter calibration of river channel migration model, the reliability of the validation data is crucial. How were the river channel location data acquired? The authors should specify if they originate from field surveys or remote sensing techniques.*

Reply: Thank you for pointing out this issue. We agree that the acquisition method and reliability of channel-position data are critical for parameter calibration of a channel-migration model. In this study, the observed river channel planform data were derived entirely from remote-sensing sources. Specifically, channel planform positions were obtained from high-resolution historical imagery available on the Google Earth platform. We selected imagery from several representative years (e.g., 2007, 2014, and 2021), which provides adequate spatial resolution and continuous coverage over the

study area, allowing the channel centerline to be clearly identified.

For data processing, we first digitized (vectorized) the channel centerline in Google Earth and exported it as a KMZ file. The KMZ data were then imported into ArcGIS, where they were transformed into shapefile format and processed under a unified coordinate reference system. We further performed geometric correction, smoothing, and topology checks to ensure the consistency of channel geometry and spatial accuracy. The resulting planform coordinates were subsequently used for model calibration and validation.

We have added a detailed description of the channel-position data acquisition and processing workflow in Section 3.3 of the revised manuscript.

5. *Figure 11 illustrates that the distribution of channel sections is narrow in the upstream areas of the basin, whereas it widens in the downstream areas. An explanation for this spatial pattern is required.*

Reply: We thank the reviewer for pointing out this issue. In our study basin, the upstream area is mountainous, whereas the downstream area is a lowland plain. The relatively narrow migration envelope (i.e., lower uncertainty) in the upstream reach is primarily because this segment is confined within a canyon setting where the valley is topographically narrow and comparatively stable. As a result, the channel is strongly constrained laterally, and the channel-migration model exhibits lower predictive uncertainty in this reach, leading to a narrower simulated distribution of channel positions.

In contrast, parts of the downstream reach located on the plain show substantially larger predictive uncertainty. This is because the low-relief terrain provides fewer topographic constraints on lateral migration, and channel behavior is more strongly influenced by the combined effects of deposition, changes in flow hydraulics, and human activities. Consequently, the basin-scale river channel migration model is associated with greater uncertainty in the downstream plain, yielding a wider predicted channel distribution, indicating higher predictive uncertainty.

**Minor Comments**

*1. To enhance the readability of the manuscript, it is advisable to add brief explanations for the technical terms, such as "Markov chain Monte Carlo" and "surrogate model".*

Reply: We have revised and improved the corresponding section in the revised manuscript.

*2. The use of "Equation" and "Eq." in the manuscript should follow a consistent format. In line 264, Equation (10) is missing a closing parenthesis.*

*The "H" in the first column of Table 6.*

Reply: We have revised and improved the corresponding section in the revised manuscript.

**Reference:**

Lei, T. L. and Lei, Z.: Harmonizing full and partial matching in geospatial conflation: a unified optimization model, ISPRS Int. J. Geo-Inf., 11, 375, https://doi.org/10.3390/ijgi11070375, 2022.

Bogoya, J. M., Vargas, A., and Schütze, O.: The Averaged Hausdorff Distances in Multi-Objective Optimization: A Review, Mathematics, 7, 894, https://doi.org/10.3390/math7100894, 2019.

Ranacher, P. and Tzavella, K.: How to compare movement? A review of physical movement similarity measures in geographic information science and beyond, Cartogr. Geogr. Inf. Sci., 41, 286–307, doi:10.1080/15230406.2014.890071, 2014.

Schütze, O., Esquivel, X., Lara, A., and Coello, C. A. C.: Using the averaged Hausdorff distance as a performance measure in evolutionary multiobjective optimization, IEEE Trans. Evol. Comput., 16, 504–522, https://doi.org/10.1109/TEVC.2011.2161872, 2012.

Bogoya, J. M., Vargas, A., and Schütze, O.: The Averaged Hausdorff Distances in Multi-Objective Optimization: A Review, Mathematics, 7, 894, https://doi.org/10.3390/math7100894, 2019.